# Nucleosome acidic patch-targeting binuclear ruthenium compounds induce aberrant chromatin condensation

Gabriela E. Davey[1], Zenita Adhireksan[1], Zhujun Ma[1], Tina Riedel[2], Deepti Sharma [1], Sivaraman Padavattan[1], Daniela Rhodes[1,3,4], Alexander Ludwig [1], Sara Sandin[1,3], Benjamin S. Murray [5], Paul J. Dyson[2] & Curt A. Davey[1,3]

The 'acidic patch' is a highly electronegative cleft on the histone H2A–H2B dimer in the nucleosome. It is a fundamental motif for protein binding and chromatin dynamics, but the cellular impact of targeting this potentially therapeutic site with exogenous molecules remains unclear. Here, we characterize a family of binuclear ruthenium compounds that selectively target the nucleosome acidic patch, generating intra-nucleosomal H2A-H2B cross-links as well as inter-nucleosomal cross-links. In contrast to cisplatin or the progenitor RAPTA-C anticancer drugs, the binuclear agents neither arrest specific cell cycle phases nor elicit DNA damage response, but rather induce an irreversible, anomalous state of condensed chromatin that ultimately results in apoptosis. In vitro, the compounds induce misfolding of chromatin fibre and block the binding of the regulator of chromatin condensation 1 (RCC1) acidic patch-binding protein. This family of chromatin-modifying molecules has potential for applications in drug development and as tools for chromatin research.

[1] School of Biological Sciences, Nanyang Technological University, 60 Nanyang Drive, Singapore 637551, Singapore. [2] Institut des Sciences et Ingénierie Chimiques, Ecole Polytechnique Fédérale de Lausanne (EPFL), CH-1015 Lausanne, Switzerland. [3] NTU Institute of Structural Biology, Nanyang Technological University, 59 Nanyang Drive, Singapore 636921, Singapore. [4] Lee Kong Chian School of Medicine, Nanyang Technological University, 59 Nanyang Drive, Singapore 636921, Singapore. [5] School of Mathematics and Physical Sciences, University of Hull, Hull HU6 7RX, UK. Gabriela E. Davey, Zenita Adhireksan, Zhujun Ma and Tina Riedel contributed equally to this work. Correspondence and requests for materials should be addressed to B.S.M. (email: b.s.murray@hull.ac.uk) or to P.J.D. (email: paul.dyson@epfl.ch) or to C.A.D. (email: davey@ntu.edu.sg)

Histone proteins package eukaryotic DNA into chromatin, yielding a structural hierarchy, in which nucleosomes comprise the basic repeating units[1]. Each nucleosome consists of a core region, composed of ~146 bp wrapped around a histone octamer, which comprises two copies each of four different core histone proteins. The H3–H4 histone tetramer organizes largely the central DNA of the nucleosome core, while the two H2A–H2B histone dimers organize the outer DNA regions. Outside of the core region, each nucleosome harbours a variable length of linker DNA, typically 10–90 bp, which can be associated with a fifth type of histone protein, linker histone.

There are at least six distinct epigenetic features of chromatin that are modulated in a site-specific and cell state-dependent manner to achieve precise control of genomic activities, notably transcription. These entail mostly attributes that relate to individual nucleosomes, including histone octamer occupancy/positioning on the double helix and an enormity of potential post-translational modifications to the histone proteins, in addition to substitutions with different histone variants and methylation of the DNA[2, 3]. Moreover, histone composition and DNA sequence can also impact nucleosome stability and dynamics properties[4, 5]. An additional epigenetic feature of chromatin regulatory structure entails consolidation of nucleosomes into higher states of compaction through linker histone association.

The pronounced changes in the gene expression profiles that yield different cell types and states means that there are a multitude of epigenetic distinctions in chromatin, which underlie disease and differentiation. In cancer cells for instance, aberrant nucleosome occupancy, histone post-translational modifications and DNA methylation work in unison to silence tumour suppressor genes while activating other genes that enable tumour development and progression[2, 3]. This suggests the possibility that compounds capable of recognizing distinguishing structural or chemical features of chromatin could allow the targeting of vulnerable points in cancer cells[6]. But on top of this, studying such compounds can also provide basic insights into molecular recognition and chromatin activity[7–9].

In studying metal-based anticancer agents that were initially expected to act therapeutically by forming DNA adducts, we had found that certain ruthenium and osmium compounds have in fact a preference to form adducts at defined histone protein sites in the nucleosome[10, 11]. This had prompted us to explore further the activities of histone-associating metalloagents as they can have certain advantages over purely organic compounds. Heavy metal centres allow distinct characteristics of coordination geometry, oxidation state and ligand exchange for fine-tuning reactivity and affinity properties of the compound[12], and whether one is focused on the structural biology, biochemistry or cellular localization, the presence of heavy atoms can make visualization and quantification more accessible.

The RAPTA $[(\eta^6$-arene)Ru(PTA)Cl$_2]$ (PTA = 1,3,5-triaza-7-phosphaadamantane) antimetastasis, antitumour and antiangiogenic compounds[13–15] have shown promise for use in chromatin research as they have a proclivity to generate protein adducts in cellular chromatin[7]. Steric access limitations to the ruthenium centre from the presence of both arene and PTA ligands disfavours DNA adduct formation[7], with RAPTA adducts forming in the nucleosome core mainly at two adjacent sites, RU1 and RU2, within a highly electronegative cleft region on the surface of the H2A–H2B dimer[9, 10]. This region, known as the acidic patch because it comprises a preponderance of glutamate/aspartate residues, coincides with a key binding platform for nuclear factor association and chromatin compaction[1, 16, 17].

Given the distinct cellular impact and molecular targeting properties of the RAPTA compounds[7, 9, 10, 13–15], here we investigate the activity of binuclear compounds that are

**Fig. 1** Structures and cytotoxicity parameters of compounds used throughout the study. Cell growth inhibition, IC$_{50}$, values (μM; HeLa cells, 40 h) are shown (mean ± s.d., $n = 3$)

composed of two RAPTA groups connected by various linker moieties, which could allow a stronger and more specific interaction with the nucleosome acidic patch. Initial characterization of binuclears synthesized with short linkers showed that their cellular cytotoxicity is much elevated over the mononuclear form and is in turn greatest with the most conformationally constrained (rigid) linkers[18]. In this work, we study these and additional binuclear agents synthesized with long linker moieties. We find that the binuclears are selective for the RU1/RU2 sites in the nucleosome acidic patch, generate substantial quantities of adducts in cellular chromatin but do not elicit a DNA damage response, and give rise to an unusual mode of cell killing associated with an irreversible state of aberrant chromatin condensation. This implies that acidic patch-targeting metalloagents could have potential for applications in chromatin studies and in the development of therapeutic compounds.

## Results

**Binuclear ruthenium compounds**. The binuclear agents were synthesized through amide-forming reactions between mononuclear ruthenium arene carboxylic acid [(toluene-$p$-propionic acid)Ru(1,3,5-triaza-7-phosphaadamantane)X] (X = Cl$_2$ or oxalate) and diamine compounds with diverse substituents linking

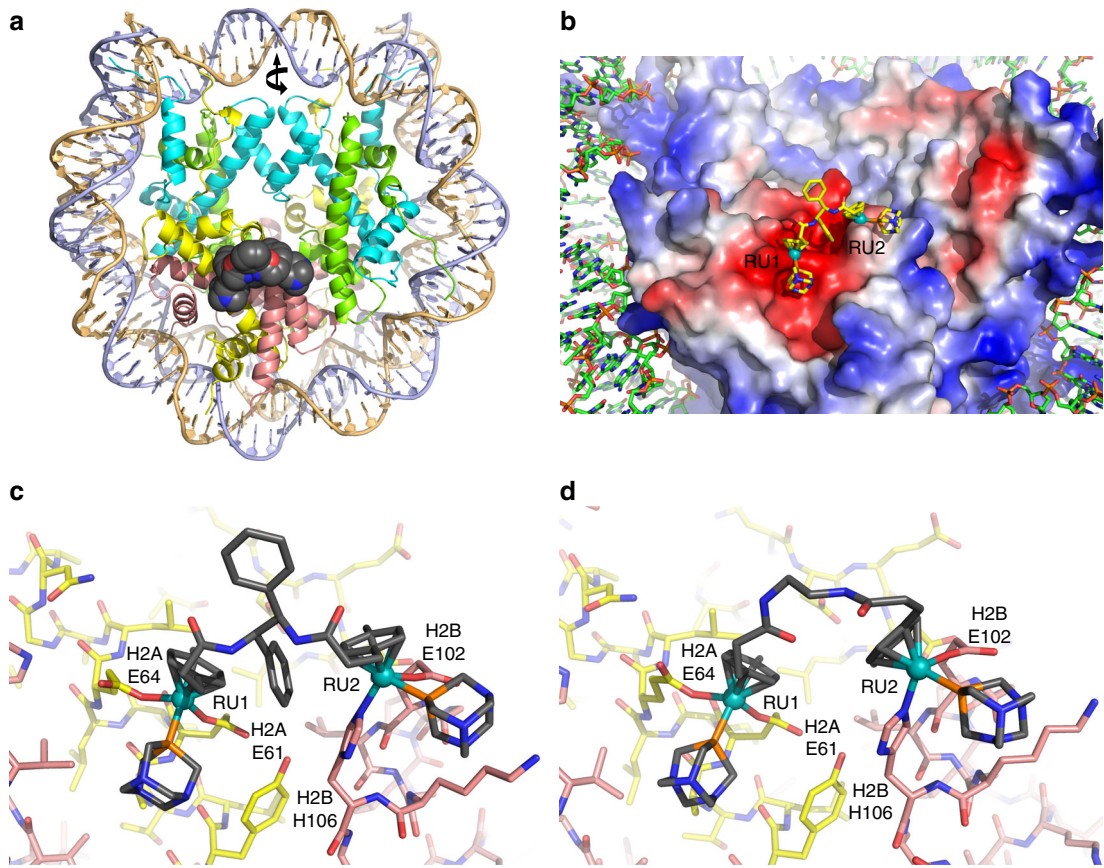

**Fig. 2** X-ray crystal structures of binuclear-treated NCP. **a** Overview of NCP with an RR adduct (space filling representation) in the acidic patch. Histone proteins are coloured cyan (H3), green (H4), yellow (H2A) and salmon (H2B), and the two DNA strands are shown in different colours. Twofold axis of pseudo-symmetry is indicated (arrow). **b** View of one face of the nucleosome core with an RR adduct. The histone octamer is rendered with an electrostatic potential surface (red, negative; blue, positive) to emphasize the acidic patch. **c**, **d** Close up views of histone interactions with the RR (**c**) or the C2 (**d**) adduct

the amine groups[18] (Fig. 1; Supplementary Figs. 1–5). We generated two distinct classes of binuclears with either rigid 1,2-diphenylethylenediamide linkers, yielding three stereochemically distinct species (RR, SS, RS), or flexible methylene or polyethylene glycol amide linkers. In the latter category, we characterized three compounds: one with a short, 2-methylene spacer (C2) and two others with long linkers, 10-unit methylene (C10) and 13-unit polyethylene glycol (PEG).

**Selective for the nucleosome acidic patch**. To assess the nucleosome site selectivity of the binuclear agents, we carried out X-ray structural characterization of nucleosome core particle (NCP) crystals incubated with each of the compounds (Supplementary Tables 1–4). This included NCP treatments with six chlorido and five oxalato compounds at different concentrations and durations. For all of the chlorido agents, substantial adduct formation is observed first at the RU1 site, entailing bifunctional coordination of the ruthenium cation by the E61 and E64 side chains of H2A (Fig. 2; Supplementary Figs. 6–8; Supplementary Table 4). For the RR, SS, C2, C10 and PEG agents, high occupancy of the RU1 site is proceeded by substantial adduct generation at the RU2 site, entailing bifunctional coordination of the ruthenium cation by the E102 and H106 side chains of H2B. This coincides with the same adduct formation profile observed for the mononuclear RAPTA drugs, wherein RU1 is favoured as the initial site of adduct formation, which in turn fosters adduct generation at the adjacent RU2 site by providing an additional hydrophobic interaction surface for the carrier ligands[9, 10]. In

contrast to the chlorido agents, none of the oxalato compounds yield evidence of adduct formation at any sites on the NCP. Therefore, the aquation of these agents is too slow under the crystal buffer conditions, rendering them unreactive, and we thus only pursued the chlorido agents further.

For the three agents with flexible linkers—C2, C10 and PEG—little additional electron density is observed beyond that corresponding to the two RAPTA groups (Supplementary Fig. 7). On the other hand, for the three compounds with rigid linkers—RR, SS and RS—there is electron density between the two RAPTA groups, consistent with the linker moiety (Supplementary Figs. 6 and 8). For the RR and SS isomers, the two RAPTA groups are positioned side by side on one face of the diphenyl linker, whereas for the RS isomer, the RAPTA groups are situated instead on opposing faces of the diphenyl linker[18]. This differential configuration allows the RR and SS agents to form bridging adducts at the RU1 and RU2 sites, thereby cross-linking the H2A and H2B histones (Fig. 2c). However, subsequent to coordination of the first ruthenium ion of the RS agent at RU1, the second ruthenium ion is correspondingly locked out of position to allow coordination at the adjacent RU2 site. Instead, the open, extended conformation of RS fosters coordination of the second ruthenium ion at a more distant histone site, the E91 side chain of H2A (Supplementary Fig. 8b).

In addition to the primary RU1 and H2A E91 sites, there is some slight degree of adduct generation by RS apparent at the RU2 site, but this must coincide with reaction of a second RS molecule, given the stereochemical constraints of the linker

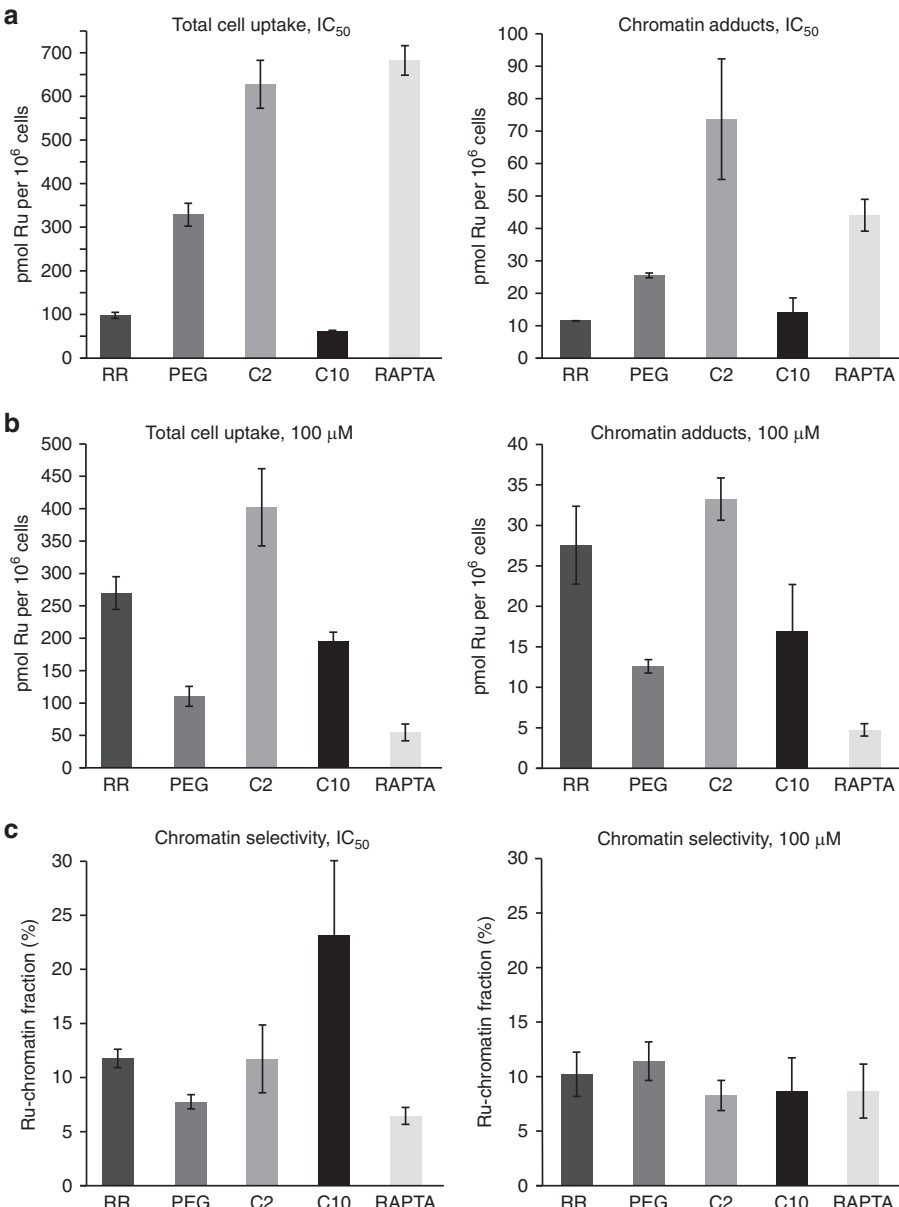

**Fig. 3** Quantification of cellular uptake and chromatin binding of binuclears and mononuclear RAPTA drug. **a**, **b** Cells were treated with binuclear agent or RAPTA-C (RAPTA) either at concentrations corresponding to their respective IC$_{50}$ values (Fig. 1; mean ± s.d., $n = 3$; **a**) or at an equimolar concentration of 100 μM (**b**; mean ± s.d., $n = 3$). Ruthenium levels were determined by inductively coupled plasma-mass spectrometry. **c** Chromatin selectivity is expressed as the fraction of the total agent taken up by the cell that is chromatin-associated (chromatin adducts/total cell uptake) for either the IC$_{50}$ (left) or 100 μM (right) treatments (mean ± s.d., $n = 3$)

(Supplementary Fig. 8b). As such, this implies that coordination of a second binuclear molecule at RU2 is sterically disfavoured subsequent to occupation of the RU1 site by the first molecule. Moreover, subsequent intramolecular chelation at RU2 should be favoured on entropic grounds, as long as this is permitted by the linker configuration. In this regard, modelling of the C2 structure shows that even the shortest linker would allow RU1-RU2 cross-linking (Fig. 2d). It is therefore likely that the majority of adducts in the crystal structures for C2, C10, PEG, RR and SS are bridging RU1-RU2 cross-links, and the lack of clarity in the connecting electron density a consequence of linker flexibility.

**Generate substantial chromatin adducts in cells**. To further analyse the targeting and impact of the binuclears, we focused on

a subset of the RU1-RU2 cross-linking agents, consisting of the one with the short flexible linker (C2), the two with the longest linkers (C10, PEG), each of which has a distinct linker chemistry, and one with a rigid linker (RR). Comparison of cell growth inhibition values (Fig. 1) shows that the four binuclears are all substantially more cytotoxic than the mononuclear RAPTA progenitor drug (RAPTA-C; C = cymene), spanning a range of about sixfold (PEG) to 64-fold (C10) greater cytotoxicity. Moreover, the most potent compounds tested, C10 and RR, are roughly as cytotoxic as the classic DNA cross-linking drug, cisplatin.

We next quantified cellular uptake and chromatin adduct formation for the different agents, which shows that these parameters are roughly proportional to the treatment

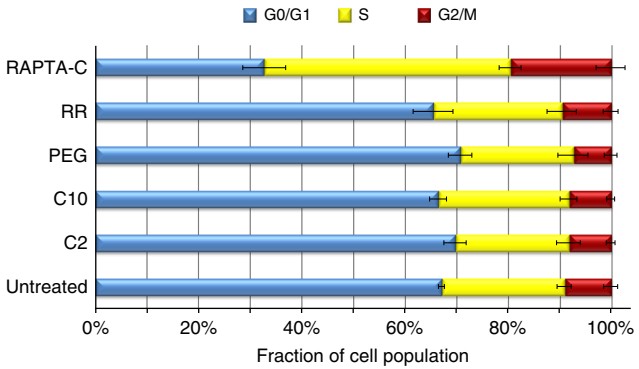

**Fig. 4** Cell cycle analysis of binuclear and mononuclear RAPTA drug-treated cells. Cell cycle profiles are based on analysis of cultured tumour cells by flow cytometry (mean ± s.d., $n = 6$)

concentration for the binuclears (Fig. 3). That is, the lower treatment concentration or $IC_{50}$ value corresponds to lesser uptake and fewer resulting chromatin adducts, and when cells are treated with equimolar agent concentrations (100 μM), roughly similar levels of uptake and chromatin binding for the binuclears are achieved. Moreover, for both the binuclears and RAPTA-C, the level of chromatin adducts formed is rather proportional to the amount of compound taken up by the cell. In contrast, however, the uptake and chromatin targeting efficiency of the four binuclears is generally much greater than that of RAPTA-C as indicated by the higher values achieved in the equimolar concentration treatments (Fig. 3b). In terms of the intracellular chromatin site selectivity, the fraction of ruthenium taken up by the cell that associates with chromatin is significantly higher for the binuclears compared to RAPTA-C for the $IC_{50}$ treatments (ranging from 8 to 23% vs. 6%), whereas this parameter is nearly constant for the equimolar treatments (spanning a narrower range of 8–11%; Fig. 3c).

**Absence of DNA damage response**. Given the greater chromatin targeting activity of the binuclears over the mononuclear RAPTA agent, we conducted cell cycle and response analysis to assess overall impact and in particular whether the binuclears generate any significant degree of DNA adducts. For the cell cycle analysis, cells were treated for 40 h at the $IC_{50}$ concentrations (Fig. 1) of the compounds to ensure subjection of a severe and uniform level of trauma. Nonetheless, the cell cycle profiles of the four binuclears are, within statistical error, identical to that of the untreated cells (Fig. 4; Supplementary Fig. 9). In contrast, the RAPTA-C-treated cells elicit a substantial degree of arrest in both the S and G2/M phases. This suggests that the binuclears do not yield significant levels of DNA adducts, as this would otherwise be expected to cause inhibition of DNA replication or transcription, resulting in stalling at S or G2/M. On the other hand, we had previously shown that a minor fraction of chromatin-associated RAPTA-C adducts pertain to DNA binding[7], which could rationalize the cell cycle impact we observe here.

To further substantiate that the binuclears do not target the DNA, we conducted western blot analysis for DNA damage markers (Supplementary Figs. 10 and 11). This indicates a slight degree of DNA damage response from RAPTA-C-treated cells relative to the strong effect stemming from cisplatin treatment. In contrast, treatment with the most cytotoxic binuclear compounds, C10 and RR, yields DNA damage signals that are no greater than that of the untreated control cells (background level).

**Induce aberrant chromatin condensation in cells**. Considering the lack of impact on the cell cycle and the absence of DNA damage response and apparent DNA targeting, in spite of cisplatin-like cytotoxicity for some of the binuclear agents, we wanted to understand how these compounds induce cell death. We set up a live cell imaging system, where cells had been stably transfected with an H2B-EGFP (H2B histone fused with enhanced green fluorescent protein) plasmid that allows visualization of chromatin. In this way, we followed nuclear activity over the course of 24 h subsequent to pre-treatment with either one of the binuclears, RAPTA-C or cisplatin (Fig. 5; Supplementary Movies 1–7). For untreated cells, division is observed to proceed in regular fashion, with the start and completion of mitosis taking place in under 1 h.

For samples treated with cisplatin, cells are typically observed to transform from a normal appearing nucleus suddenly into the apoptotic state. In the case of RAPTA-C treatment, cells appear to spend an extended time in mitosis before undergoing apoptosis. In contrast, the binuclear agents are seen to induce an irreversible, condensed state of chromatin, with a degree of compaction that appears similar to that of the untreated mitotic cells. This state of aberrant chromatin condensation persists for many hours before apoptosis.

**Induce spontaneous misfolding of chromatin fibre**. Testing the influence of the binuclear agents on chromatin dynamics in vitro could shed light on the striking nuclear impact of these compounds in inducing an irreversible condensed state of the chromatin. For this, we utilized a nucleosome array system consisting of thirty-six 177 bp nucleosomes in tandem (Supplementary Fig. 12), which serves as a model for chromatin fibre[19]. Native electrophoretic mobility shift assays indicate that adducts formed by the binuclears induce compaction (folding) of the nucleosome array, to a degree that is proportional to treatment strength (Fig. 6). That is, in spite of the added positive charge and increased molecular weight from the binuclear adducts, the treated array migrates faster than untreated array, indicative of a more compact configuration. In contrast, the opposite effect is seen for sub-nucleosomal histone-DNA assemblies (present to ensure integrity of the reconstituted array), which accumulate adducts as well, but migrate more slowly as a consequence. Cisplatin treatment of array yields little or no apparent compaction, whereas treatment with RAPTA-C does induce some extent of array folding, but only at high treatment concentration and to a reduced degree compared to the binuclears. In fact, high concentration binuclear treatment results in aggregation and precipitation of the array.

To further understand the effects of binuclear adducts on chromatin fibre, we carried out electron microscopic (EM) analysis of the array (Fig. 7). In the native state, under low ionic strength conditions, the array remains unfolded, adopting a random-coil, beads-on-a-string conformation. In the presence of divalent metal, at around 1.6 mM $Mg^{2+}$, the array achieves a state of maximal intramolecular compaction, yielding the so-called two-start helix configuration, in which nucleosomes 'zig-zag' along a left-handed axis[20].

In contrast to the behaviour of native material, binuclear-treated array adopts a highly compact configuration in the absence of any $Mg^{2+}$ (low ionic strength; Fig. 7; Supplementary Fig. 13). The degree of compaction appears similar to that of $Mg^{2+}$-folded native array, and the addition of $Mg^{2+}$ to the binuclear-treated samples does not yield any further compaction. Moreover, the compact binuclear configuration achieved is highly distinct relative to native material, being varied from one molecule to the next, overall irregular and displaying greater

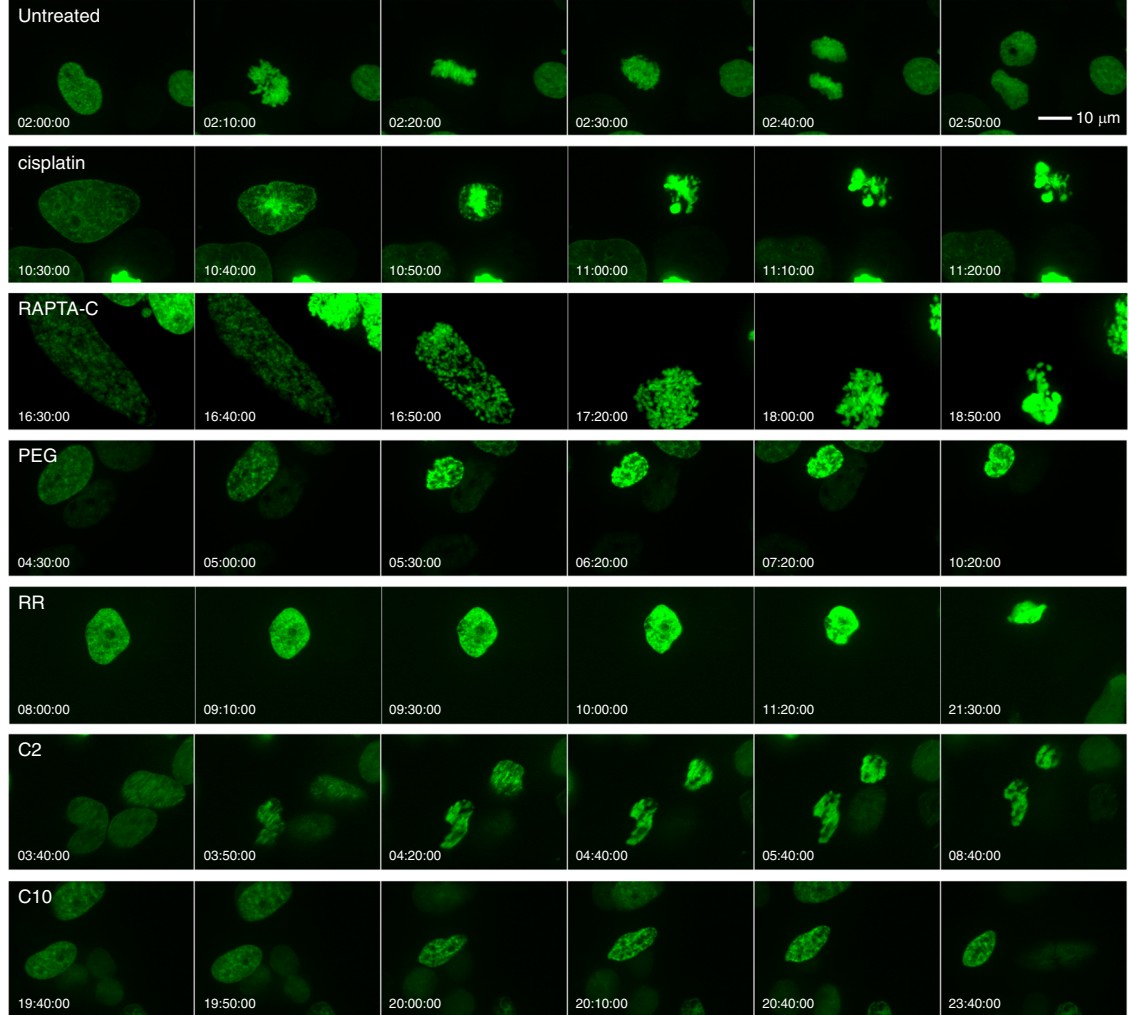

**Fig. 5** Live fluorescence imaging of drug- and binuclear-treated cells. Nuclear chromatin is visible by virtue of the incorporated H2B-EGFP histone fusion protein. Montages correspond to extractions from the 24 h imaging sequences (Supplementary Movies 1–7; times shown at bottom for each frame)

twisting and bending along the axis. Nonetheless, imaging of cisplatin- or RAPTA-C-treated array under $Mg^{2+}$-free conditions yields no pronounced compaction effect, although a slight degree of RAPTA-C-induced folding or structural perturbation is discernible (Supplementary Fig. 13).

**Impede protein binding and cross-link nucleosomes.** Since the nucleosome acidic patch is known to play a key role in nuclear factor binding and chromatin fibre folding[1, 16, 17], we investigated how the binuclear adducts may influence interactions with the nucleosome core. We tested the effect of binuclear and RAPTA-C treatments on the NCP binding of the acidic patch-associating protein, regulator of chromatin condensation 1 (RCC1)[21]. The binuclear adducts are able to inhibit or completely block the binding of RCC1, while the RAPTA-C samples, subjected to the same treatment concentrations as the binuclears, do not show binding interference (Fig. 8a). Nonetheless, at high treatment strength, RAPTA-C is able to completely block RCC1 binding to the NCP (Fig. 8b).

For the RCC1 binding analysis, we used short compound incubation times to minimize precipitation of the derivatized NCP. When NCP was subjected to a longer incubation time with the binuclears, extensive internucleosomal cross-linking is apparent, resulting in precipitation at the higher treatment

concentrations (Fig. 8c). In contrast, for the mononuclear RAPTA drug, nucleosome-nucleosome cross-linking is not observed. Consistent with this, denaturing electrophoretic gel analysis shows distinct cross-linked histone species formed by the binuclears compared to RAPTA-C (Supplementary Fig. 14), although the harsh conditions required to denature the nucleosome are likely to also alter ruthenium adducts to at least some degree.

**Discussion**
The RAPTA-based binuclear agents characterized here display the striking ability to induce a catastrophic state of chromatin condensation, which persists for many hours and ultimately coincides with cell death. The extent of condensation is similar to that of mitotic chromosomes, but nonetheless cells are not able to recover from this compacted state once attained. The in vitro and cellular analyses suggest that the phenomenon arises from the nucleosome acidic patch-targeting activity of the binuclears. Indeed, there appears to be no significant DNA binding in the cell, although the binuclears efficiently generate adducts on cellular chromatin, there is neither a measurable impact on cell cycle profile nor elicitation of a DNA damage response.

We had previously characterized RAPTA-C binding by $IC_{50}$-concentration treatment of A2780 ovarian cancer cells and

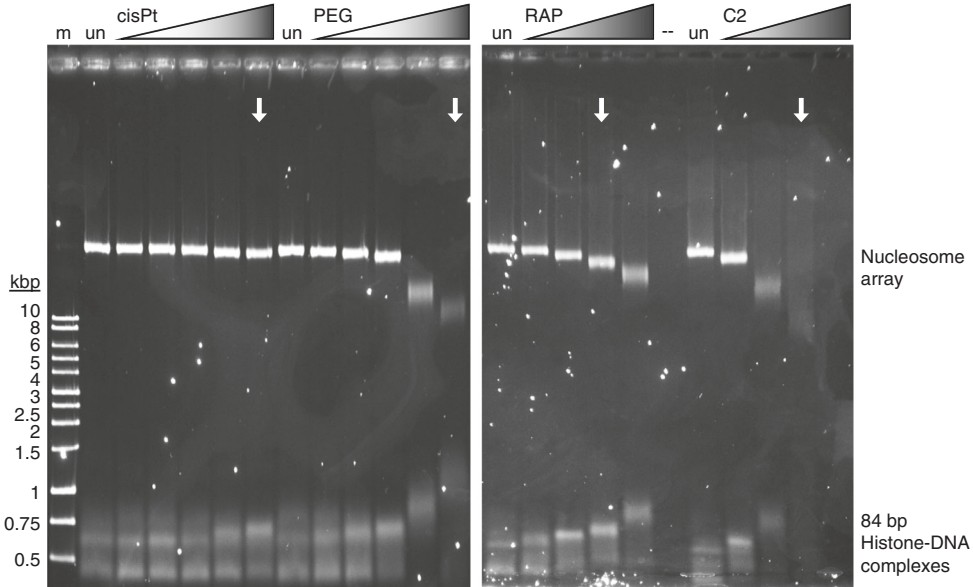

**Fig. 6** Electrophoretic mobility shift analysis of drug- and binuclear-treated nucleosome array. Agarose gel samples include native array (un) and array treated with either cisplatin (cisPt), PEG, RAPTA-C (RAP) or C2 (m, 500–10,000 bp DNA marker). Samples corresponding to equimolar high treatment concentration between the four different agents are indicated with arrows. The maximal RAPTA-C and C2 treatments are at fivefold higher concentration, at which point all of the binuclear-treated material is lost to precipitation

determined that ~4% of the intracellular ruthenium content is associated with chromatin[7, 10]. A similar degree of chromatin site selectivity is apparent for the classic DNA-targeting agent cisplatin[22]. In the present study, we observe a somewhat higher level of site preference for chromatin in HeLa cells, at respectively, 6% or 9%, depending on whether cells are treated with an $IC_{50}$ or 100 µM concentration of RAPTA-C. Substantial chromatin adduct levels would rationalize the activity observed here with the cell imaging experiments, whereby RAPTA-C is seen to induce apoptosis by interfering with the mitotic process. This is moreover consistent with the substantial degree of G2/M phase arrest caused by this drug, as seen in this study and previously[7] for different cancer cell types. Indeed, a recent investigation found that RAPTA-C induces increased formation of DNA bridges in cancer cells[23], which is consistent with its partial DNA-targeting activity[7] and could help further rationalize the distinct cellular impact we observe here. Nonetheless, the binuclear activities are decisively different from those of this progenitor mononuclear RAPTA drug and their chromatin targeting abilities are also superior, both in terms of more efficient cellular uptake and chromatin adduct formation, as well as overall greater intracellular localization to chromatin.

The crystallographic studies show that all of the binuclears, with the exception of the 'locked-out' RS[18], are capable of forming bridging adducts at sites RU1 and RU2, consequently cross-linking H2A and H2B within the dimer. On the other hand, binuclear treatments of NCP in solution yield inter-nucleosomal cross-links, which are presumably acidic patch-acidic patch (RU1–RU1; H2A–H2A) cross-linking. Although we do not observe such cross-links in the NCP crystals, the crystal packing configuration and site-blocking inter-particle contacts would necessitate a binuclear linker length on the order of 100 Å or so to allow these to form (the fully extended conformation of our longest binuclear, PEG, coincides with a Ru-Ru separation of ~32 Å). Nonetheless in the solution state, where nucleosomal interactions are dynamic, even the shortest binuclear agents, RR and C2, are effective at cross-linking nucleosomes. Since this is a

property that is not observed for the progenitor mononuclear RAPTA drug, it may rationalize the higher cytotoxicities of the binuclears and moreover contribute to their nucleosome array misfolding activity in vitro and anomalous chromatin condensing activity in the cell. At the same time, however, the broad range of cytotoxicity displayed by PEG, C2, RR and C10 indicates an important role of linker flexibility[18], length and also possibly chemical nature, in modulating cellular impact. It is interesting to speculate that these features may influence chromatin binding and cross-linking activity in a structure-/site-specific fashion to yield differential cytotoxic effects. However, at present we cannot rule out the possibility of other functionally important cellular protein targets. In any case, we show here that the binuclear and RAPTA-C adducts are also capable of inhibiting the binding of RCC1 and seemingly other acidic patch-binding nuclear proteins[9, 16], and this is likely to contribute to their cellular impact.

In addition to the cross-linking activity, another feature of acidic patch association that likely contributes to the chromatin compacting activity of the binuclears relates to electrostatic and surface contour effects of the adducts. Indeed, we observe a modest degree of nucleosome array folding ability with the mononuclear RAPTA drug, which does not show any nucleosome-nucleosome cross-linking activity. Studies have demonstrated that the acidic patch plays a key function in chromatin fibre compaction, and the binding of protein factors to this region can modulate chromatin structure and dynamics[17, 24, 25]. In particular, the effect of the binuclears appears to be analogous to that of the acidic patch-targeting viral peptide, LANA[26]. The 23-amino acid LANA peptide promotes $Mg^{2+}$-induced folding and aggregation of nucleosome array and contributes a 4+ charge[24] to the acidic patch, like the binuclear adducts. Moreover, the LANA peptide also alters nuclear architecture and chromatin condensation in cells. Nonetheless, unlike the LANA peptide, the binuclears are capable of inducing seemingly full, albeit aberrant, compaction in vitro under $Mg^{2+}$-free conditions and bring about dramatic condensation in cells. This likely arises from the added

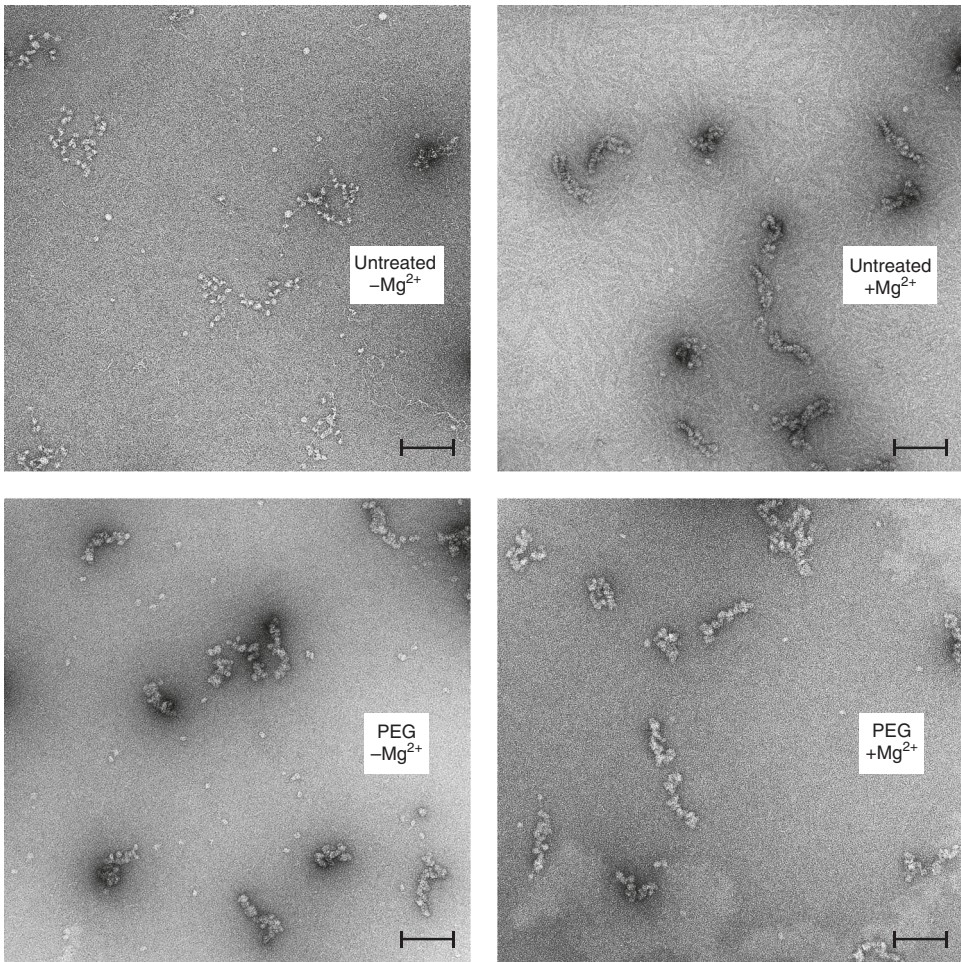

**Fig. 7** Electron microscopy of negatively stained native and binuclear-treated nucleosome array. Samples include native array (untreated) with and without 1.6 mM $Mg^{2+}$ and PEG-treated array with and without 0.5 mM $Mg^{2+}$ (binuclear-treated array aggregates/precipitates at higher $Mg^{2+}$ concentrations). The black scale bar inset corresponds to 100 nm

nucleosome cross-linking properties of the binuclears, since a similar electrostatic impact could be achieved by either a single LANA peptide molecule or saturating levels of RAPTA-C that introduce the same 4+ charge into the acidic patch. Moreover, the impact of peptide-acidic patch association can itself be varied, as the cytomegalovirus IE1 C-terminal domain peptide has in fact been shown to inhibit linker histone-induced folding of nucleosome array[27]. In this case, however, the peptide has a lower net charge (3+) and binds the acidic patch with an aspartate and carboxy-terminal element protruding into solvent space, which could disfavour inter-nucleosomal interactions.

There are few previously reported exogenous ligands that are known to target the acidic patch, including just several different types of related Ru/Os-arene compounds[7, 9–11, 28] and the naturally occurring viral peptides. Compared to the viral peptides, which like the binuclears are also polyvalent, the binuclears have unprecedented activities in controlling chromatin structure and compaction. These properties could potentially be exploited for developing chromatin investigation tools or therapeutic agents. For instance, it may be possible to design binuclears that are selective for specific H2A variants, which differ with respect to acidic patch charge distribution or structure. This could allow for targeting particular chromatin sites and activity states. Moreover, binuclear adducts are likely to accumulate in the nucleus at sites where the acidic patch is unoccupied by protein factors. In the NCP crystals, we observe adduct formation at just one of the two symmetry-related acidic patches, as the other is effectively blocked by a crystal contact involving the H4 N-terminal tail. This type of interaction has been implicated in chromatin higher order structure[17, 29–31], suggesting that the H4 tail may sterically block compounds from binding to the acidic patch in condensed chromatin states, such as with silenced genes. In this way, the binuclears may preferentially associate with open states of chromatin, in particular activated genes, although more work is required to test this possibility. Conversely, once the acidic patch is occupied with a binuclear adduct, it may block the binding or alter the association mode of the H4 tail, which may underlie, at least in part, the non-native structures observed for the binuclear-compacted nucleosome array. Collectively, this suggests that compounds in this binuclear ruthenium family could help to shed light on cell type- and state-dependent distinctions in chromatin structure and activity as well as provide possibilities for differentially targeting the distinct chromatin landscape of cancer cells.

## Methods

**Binuclear compound synthesis**. *Reagents:* All chemicals were used as received from suppliers. 4,7,10-trioxa-1,13-tridecanediamine (97%) and 1,10-diaminodecane (97%), *O*-(benzotriazol-1-yl)-*N,N,N′,N′*-tetramethyluronium tetrafluoroborate (97%), *N,N*-diisopropylethylamine (99%) and acetyl chloride (98%) were purchased from Sigma-Aldrich. *N,N*-dimethylformamide (99.8%, extra dry, Acroseal®), acetonitrile (99.9%, extra dry, Acroseal®) and acetone (99.8%, extra dry, Acroseal®) were obtained from Acros Organics, and methanol (anhydrous, 99.8%) was purchased from Sigma-Aldrich.

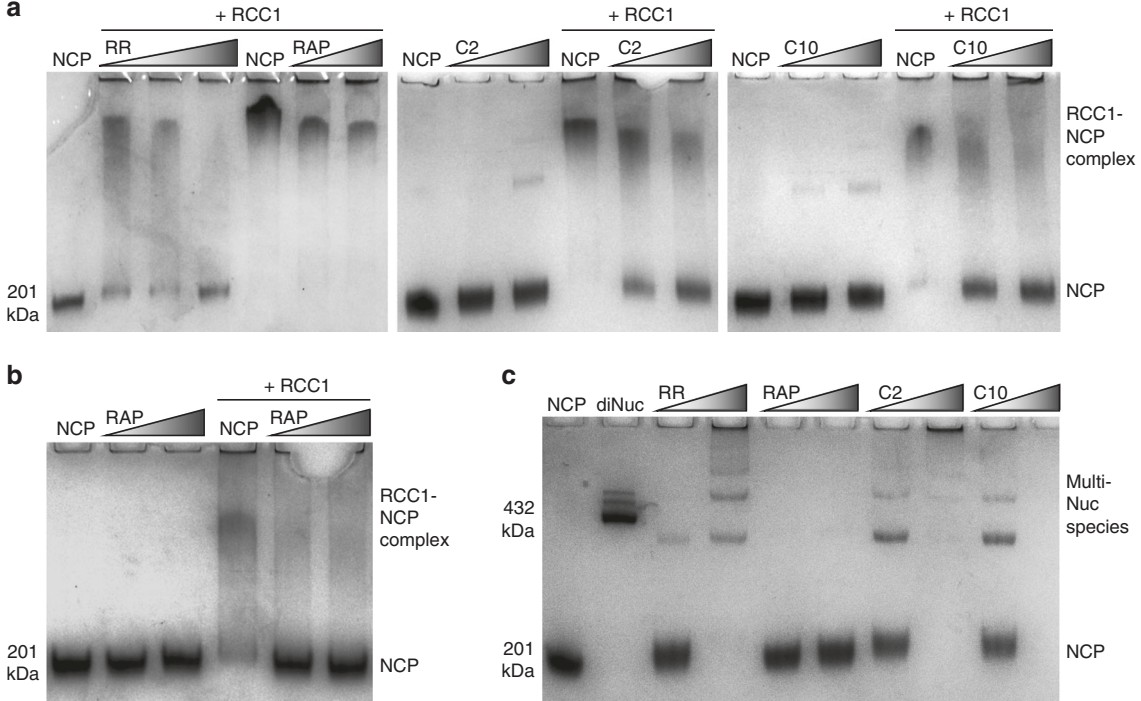

**Fig. 8** Nuclear protein binding and inter-nucleosomal cross-linking analysis of agent-treated nucleosome core particle. **a**, **b** Native PAGE RCC1-NCP binding analysis including treated (high-concentration RAPTA-C [RAP] treatment, **b**) and untreated (NCP) samples, with and without added RCC1 protein. **c** Native PAGE NCP-NCP cross-linking analysis, including untreated NCP and dinucleosome (diNuc) samples for reference. Note the higher concentration C10-treated sample is lost to precipitation

*NMR analysis:* $^{1}$H, $^{13}$C and $^{31}$P nuclear magnetic resonance (NMR) spectra were recorded on a Bruker Avance II 400 spectrometer ($^{1}$H at 400 MHz, $^{13}$C at 101 MHz and $^{31}$P at 162 MHz). Spectra are referenced internally to residual solvent peaks (D$_2$O: $^{1}$H $\delta$ 4.79, $^{13}$C $\delta$ unreferenced; DMSO-$d6$: $^{1}$H $\delta$ 2.50, $^{13}$C $\delta$ 39.52); $^{31}$P NMR spectra are reported relative to an 85% H$_3$PO$_4$ external reference.

*Mass spectrometry analysis:* A Q-Tof (Quadrupole Time-Of-Flight) Ultima mass spectrometer (Waters) was utilized in the acquisition of electrospray-ionization mass spectrometry (ESI-MS) data. The spectrometer was fitted with a Z-spray source and LockSpray interface and was operated in positive ionization mode. Sample solutions were made in CH$_3$CN:H$_2$O:HCOOH (50:49.9:0.1) ($\sim$10$^{-5}$ M) and were infused into the spectrometer (20 μl per min) with a mixture of CH$_3$CN:H$_2$O:HCOOH (50:49.9:0.1). The capillary voltage was set at 3.5 kV, the sample cone set at 35 V, the source temperature, and the desolvation temperatures were set at 80 °C and 200 °C, respectively, and the acquisition window spanned $m/z$ 300–1500 in 1 s. A phosphoric acid solution (0.01%), introduced via an orthogonal ESI probe, was utilized as an external calibrant. A correction factor for the mass scale was determined from LockSpray data to provide accurate mass information of the analyte. MassLynx 4.1 software was used for data processing.

*Elemental analysis:* Elemental analysis was carried out using an EA1108 Elemental Analyzer (Fisons Instruments).

*Synthesis overview:* The overall approach for synthesizing the binuclear agents, shown in Supplementary Fig. 15, is based on our previously reported method for producing the chlorido and oxalato derivatives of C2, RR, SS and RS[18]. Below, we describe the synthesis and analysis of the chlorido (Cl4) and oxalato (Ox) derivatives of PEG and C10.

*RuPEGOx:* Ruthenium acid monomer (300 mg, 0.588 mmol) and *O*-(benzotriazol-1-yl)-*N,N,N′,N′*-tetramethyluronium tetrafluoroborate (TBTU; 188 mg, 0.586 mmol) were suspended in *N,N*-dimethylformamide (DMF; 2.5 ml) followed by the addition of *N,N*-diisopropylethylamine (DIPEA; 102 μl, 0.586 mmol). The yellow suspension was stirred for 10 m followed by the addition of 4,7,10-trioxa-1,13-tridecanediamine (58 μl, 0.265 mmol). The mixture was stirred for 2 h under an atmosphere of N$_2$ and then filtered to isolate unreacted ruthenium monomer acid. The collected yellow filtrate was added dropwise to acetone (15 ml) to yield a precipitate, and the suspension was diluted by the addition of MeCN (10 ml) and then sonicated for 5 min. The precipitate was isolated by filtration to yield a hydroscopic yellow solid, which was dried under high vacuum to yield the crude product as a yellow hydroscopic solid that was utilized without further purification (200 mg, 0.166 mmol, 63%). $^{1}$H NMR (D$_2$O, 400 MHz): $\delta$ = 5.92–5.99 (m, 8H, Ar), 4.60 (s, 12H, PTA), 4.18 (s, 12H, PTA), 3.58–3.73 (m, 8H, O–CH$_2$–CH$_2$–O), 3.47 (t, $J$ = 6.5 Hz, 4H, 2 × CH$_2$), 3.21 (t, $J$ = 6.5 Hz, 4H, 2 × CH$_2$), 2.61 (s, 8H, 4 × CH$_2$), 2.08 (s, 6H, CH$_3$), 1.71 (p, $J$ = 6.5 Hz, 4H, 2 × CH$_2$–CH$_2$–CH$_2$); $^{31}$P{$^{1}$H} NMR

(D$_2$O, 162 MHz): $\delta$ = −33.2; $^{13}$C{$^{1}$H} NMR (D$_2$O, 101 MHz): $\delta$ = 173.8 (2C, amide C=O), 166.1 (4C, oxalate C=O), 98.9 (2C, Ar$_{(q)}$), 96.6 (2C, Ar$_{(q)}$), 88.4 (d, 4C, $J$ = 3.5 Hz, Ar), 87.7 (d, 4C, $J$ = 4.0 Hz, Ar), 70.7 (d, 6C, $J$ = 7.0 Hz, PTA), 69.6, 69.4, 68.3 (6C, 6 × CH$_2$), 48.6 (d, 6C, $J$ = 15.5 Hz, PTA), 36.4, 35.6, 28.3 (8C, 8 × CH$_2$), 17.4 (2C, 2 × CH$_3$); high resolution mass spectrometry (HRMS) (ES$^{+}$) $m/z$ found 604.1315 [M + 2H]$^{2+}$ C$_{46}$H$_{70}$N$_8$O$_{13}$P$_2$Ru$_2$ requires 604.1325. See Supplementary Fig. 2 for NMR spectral profiles.

*RuPEGCl4 (PEG):* RuPEGCl4 was synthesized using reported methodology via RuPEGOx[18]. This entailed slow addition of acetyl chloride (890 μl, 12.5 mmol) to MeOH (5 ml) at 0 °C under an atmosphere of N$_2$. The solution was allowed to stir for 10 m followed by the addition of RuPEGOx (80 mg, 0.066 mmol) as a solution in MeOH (2 ml). A red precipitate instantly formed; the mixture was further stirred for 16 h. The solvent was then decanted and the oily red residue was washed with MeOH (2 × 5 ml) and then dried under reduced pressure to yield the tetra-hydrochloride salt of the desired product as a hydroscopic red solid (40 mg, 0.030 mmol, 45%). $^{1}$H NMR (DMSO-$d6$, 400 MHz): $\delta$ = 7.93 (s br, 2H, amide NH), 5.82–5.96 (m br, 8H, Ar), 4.70–4.93 (m br, 12H, PTA), 4.29 (s br, 12H, PTA), 3.41–3.53 (m, 8H, –O–CH$_2$–CH$_2$–O–), 3.33 (m, 4H, 2 × CH$_2$), 3.05 (m, 4H, 2 × CH$_2$), 2.37 (s, 8H, 2 × –ArCH$_2$CH$_2$CO–), 1.91 (s, 6H, 2 × CH$_3$), 1.58 (m, 4H, 2 × CH$_2$–CH$_2$–CH$_2$); $^{31}$P{$^{1}$H} NMR (DMSO-$d6$, 162 MHz): $\delta$ = −26.7; $^{13}$C{$^{1}$H} NMR (DMSO-$d6$, 101 MHz): $\delta$ = 170.5 (2C, amide C=O), 97.8, 97.4 (4C, Ar$_{(q)}$), 88.4 (d, 4C, $J$ = 5.0 Hz, Ar), 87.8 (d, 4C, $J$ = 4.5 Hz, Ar), 70.2 (br, 6C, PTA), 69.7, 69.5, 68.0 (6C, 6 × –CH$_2$–O–), 48.2 (d, 6C, $J$ = 19.5 Hz, PTA), 35.8, 35.1 29.3, 28.2 (8C, 8 × CH$_2$), 17.9 (2C, Ar–CH$_3$); HRMS (ES$^{+}$) $m/z$ found 1173.1698 [M + H]$^{+}$ C$_{42}$H$_{69}$Cl$_4$N$_8$O$_5$P$_2$Ru$_2$ requires 1173.1703; C$_{42}$H$_{68}$Cl$_4$N$_8$O$_5$P$_2$Ru$_2$•4HCl (%): calcd C 38.31 H 5.51 N 8.51; found C 38.43 H 5.55 N 8.36. See Supplementary Fig. 3 for NMR spectral profiles.

*RuC100x:* Ruthenium acid monomer (200 mg, 0.392 mmol) was suspended in DMF (1.5 ml) under an atmosphere of N$_2$ followed by the addition of DIPEA (137 μl, 0.787 mmol) and TBTU (126 mg, 0.392 mmol). The suspension was stirred for 10 m followed by the addition of 1,10-diaminodecane (16.9 mg, 0.098 mmol) and left to stir for 2 h. The yellow suspension was then subjected to centrifugation to isolate the crude product as a solution in DMF. The DMF layer was then added dropwise to acetone (12 ml) to precipitate a yellow powder that was further washed with acetone (10 ml) to yield the desired product that was used without further purification (80 mg, 0.069 mmol, 70 %). $^{1}$H NMR (D$_2$O, 400 MHz): $\delta$ = 5.94 (m br, 8H, Ar), 4.58 (s, 12H, PTA), 4.17 (s, 12H, PTA), 3.12 (t, $J$ = 6.5 Hz, 4H, CH$_2$–NH), 2.60 (s br, 8H, CH$_2$–CH$_2$–CO), 2.07 (s, 6H, CH$_3$), 1.39 (p, $J$ = 7.0 Hz, 4H, CH$_2$–CH$_2$–NH), 1.08–1.25 (m, 12H, alkyl CH$_2$); $^{31}$P{$^{1}$H} NMR (D$_2$O, 162 MHz): $\delta$ = −33.4; $^{13}$C{$^{1}$H} NMR (D$_2$O, 101 MHz): $\delta$ = 173.6 (2C, amide C=O), 166.0 (4C, oxalate C=O), 99.0 (2C, Ar$_{(q)}$), 96.2 (2C, Ar$_{(q)}$), 88.6 (d, $J$ = 3.5 Hz, 4C, Ar), 87.7 (d,

$J = 4.0$ Hz, 4C, Ar), 70.7 (d, $J = 7.0$ Hz, 6C, PTA), 48.5 (d, $J = 15.5$ Hz, 6C, PTA), 39.3, 35.8, 28.7, 2 x 28.3, 28.2, 26.0 (14C, $14 \times CH_2$), 17.4 (2C, $2 \times CH_3$); HRMS (ES$^+$) $m/z$ found 580.1384 $[M + 2H]^{2+}$ $C_{46}H_{70}N_8O_{10}P_2Ru_2$ requires 580.1401. See Supplementary Fig. 4 for NMR spectral profiles.

*RuC10Cl4 (C10)*: RuC10Cl4 was synthesized using previously described methodology via RuC10Ox[18]. This entailed slow addition of acetyl chloride (890 µl, 12.5 mmol) to MeOH (5 ml) at 0 °C under an atmosphere of N$_2$. The solution was allowed to stir for 10 m followed by the addition of RuC10Ox (68 mg, 0.059 mmol) as a suspension in MeOH (2 ml). A red precipitate instantly formed; the mixture was stirred for 16 h. The solvent was then decanted and the red residue was washed with MeOH ($2 \times 5$ ml) then dried under reduced pressure to yield the hydrochloride salt of the desired product as a red solid (29 mg, 0.023 mmol, 39%). $^1$H NMR (DMSO-$d6$, 400 MHz): $\delta = 7.90$ (br, 2H, amide NH), 5.81–5.94 (m, 8H, Ar), 4.69–4.92 (m, 12H, PTA), 4.29 (s, 12H, PTA), 2.98 (m, 4H, $2 \times -CH_2-NH-$), 2.37 (s, 8H, $2 \times -CH_2-CH_2-CO-$), 1.91 (s, 6H, $2 \times -CH_3$), 1.33 (m, 4H, $-CH_2-CH_2-NH-$), 1.13–1.24 (m, 12H, 6 x $CH_2$); $^{31}$P{$^1$H} NMR (DMSO-$d6$, 162 MHz): $\delta = -26.7$; $^{13}$C{$^1$H} NMR (DMSO-$d6$, 101 MHz): $\delta = 170.3$ (2C, amide C=O), $2 \times 97.8$ (4C, Ar$_{(q)}$), 88.4 (d, $J = 4.0$ Hz, 4C, Ar), 87.7 (d, $J = 5.0$ Hz, 4C, Ar), 70.2 (6C, PTA), 48.2 (6C, PTA), 38.4, 35.2, 29.1, 29.0, 28.7, 28.2, 26.4 (14C, $14 \times CH_2$), 17.9 (2C, $2 \times CH_3$); HRMS (ES$^+$) $m/z$ found 1125.1792 $[M + H]^+$ $C_{42}H_{69}Cl_4N_8O_2P_2Ru_2$ requires 1125.1855; $C_{42}H_{68}Cl_4N_8O_2P_2Ru_2\bullet4HCl$ (%): calcd C 39.76 H 5.72 N 8.83; found C 40.00 H 5.74 N 8.53. See Supplementary Fig. 5 for NMR spectral profiles.

**Crystallographic analysis of nucleosome core particle**. X-ray crystallographic analysis was conducted using NCP assembled with recombinant *Xenopus laevis* or *Homo sapiens* histones and a 145 bp DNA fragment[32]. *H. sapiens* core histone expression plasmids[33] were kindly provided by Hitoshi Kurumizaka (Waseda University, Japan) and Thirumananseri Kumarevel (RIKEN Harima Institute at SPring8, Japan). The hanging droplet method was used to grow NCP crystals from buffers containing MnCl$_2$, KCl and K-cacodylate [pH 6.0][34]. Crystals were harvested and transferred into a stabilization buffer (37 mM MnCl$_2$, 40 mM KCl, 20 mM K-cacodylate [pH 6.0], 24% 2-methyl-2,4-pentanediol and 2% trehalose). MgSO$_4$ was substituted in place of MnCl$_2$ by thorough rinsing of crystals with a magnesium buffer (10 mM MgSO$_4$, 20 mM K-cacodylate [pH 6.0], 24% 2-methyl-2,4-pentanediol and 2% trehalose)[10].

To obtain NCP-binuclear adduct structures, native NCP crystals were subjected to 15- to 67-h incubation with magnesium buffer containing 1 or 2 mM binuclear agent (Supplementary Tables 1–4). Treated crystals were mounted directly into a cryocooling N$_2$ gas stream set at −175 °C[7]. X-ray diffraction data were recorded at beam line X06DA of the Swiss Light Source (Paul Scherrer Institute, Villigen, Switzerland) at an X-ray wavelength of 1.50 Å with a Pilatus 2M-F detector. Data were processed using MOSFLM[35] and SCALA from the CCP4 suite[36].

Initial solution of NCP-binuclear adduct structures was achieved by molecular replacement using the crystal structure of NCP containing RAPTA-C adducts (pdb code 3MNN)[10] as the reference model. Routines from the CCP4 suite[36] were used to conduct structural refinement and model building. Small molecule crystal structures of the oxalato derivatives of C2, RR, SS and RS[18] were used to compose stereochemical restraint parameters for the binuclear adducts. Data collection and structural refinement statistics are given in Supplementary Tables 1–4. Molecular graphics images were produced with PyMOL (DeLano Scientific LLC, San Carlos, CA, USA).

**Determination of cell growth inhibition parameters**. The human cervical carcinoma cell line, HeLa, was a gift from Peter Dröge (NTU, Singapore). HeLa cells were grown at 37 °C and 5% CO$_2$ in DMEM medium containing 10% foetal calf serum with 2 mM glutamine, 100 units per ml penicillin and 100 µg ml$^{-1}$ streptomycin. To measure the cytotoxicity from exposure to different agents, cells were seeded in 96-well plates (5000 cells per well) and grown for 24 h. Stock solutions were prepared by dissolving cisplatin (purchased from Sigma-Aldrich, USA (P4394-250MG)), RAPTA-C, C2, C10, PEG or RR in complete medium. These medium stocks were then subjected to serial dilutions and added to the cells at various concentrations. Media alone was added to the untreated control cells.

Following a 72 h (IC$_{50}$ values used for DNA damage analysis) or 40 h (IC$_{50}$ values used for other experiments) incubation, media were aspirated and 100 µl of 10% 3-(4,5-dimethylthiazol-2-yl)-2,5-diphenyltetrazolium bromide in DMEM complete medium (TOX1-1KT; MTT kit, Sigma-Aldrich) was added to cells, which were incubated for 3 h at 37 °C. Subsequently, 100 µl of solubilization buffer was added to each well with vigorous pipetting in order to dissolve the formazan. The resulting optical density was measured at 570 and 690 nm using a multi-well plate reader (Infinite M200 PRO, Magellan data analysis software, TECAN, Switzerland). The ratios of surviving cells were calculated by comparing to the untreated samples, and the IC$_{50}$ was derived based on at least three independent measurements.

**Binuclear uptake and chromatin adduct quantification**. *Cell culture and binuclear treatment:* HeLa cells, obtained from the European Centre of Cell Cultures (ECACC, Salisbury, UK), were kindly provided by Claudia Battistella (School of Engineering, EPFL, CH). The cells were maintained in DMEM GlutaMax medium supplemented with 10% foetal bovine serum and 1% penicillin/streptomycin in a humidified environment at 37 °C with 5% CO$_2$. For total cell uptake and chromatin

binding experiments, cells were grown as adherent monolayers in six-well plates or 75 cm$^2$ flasks, respectively, for 24 h prior to drug exposure. Then the cells were incubated at 37 °C for 24 h with the binuclear compounds in complete culture medium at a concentration of 100 µM or at the respective IC$_{50}$ concentration (Fig. 1). Subsequently, the cells were washed twice with phosphate buffered saline solution to remove unbound drug and harvested by using an enzyme-free cell dissociation buffer (Millipore, Switzerland) and pelleted by centrifugation at $100\times g$ for 4 min. The experiments were performed in triplicate.

*Chromatin isolation:* Prior to chromatin isolation, a cross-linking reaction using 1% formaldehyde in PBS was performed for 10 min at room temperature and subsequently quenched by adding 2 M glycine (final concentration of 100 mM) for 5 min. The chromatin was extracted using a Pierce Chromatin Prep Module (Thermo Fisher Scientific, Switzerland) according to the manufacturer's protocol.

*DNA and protein quantification:* All chromatin samples were analysed for their chromosomal DNA content prior to inductively coupled plasma-mass spectrometry (ICP-MS) measurements. DNA was quantified by ultraviolet absorption measurements at 260 nm and with the PicoGreen dsDNA quantitation assay (Invitrogen). PicoGreen (50 µl per well, 200× diluted in 10 mM Tris + 1 mM EDTA) was added to 50 µl of DNA sample and the fluorescence signal was determined by spectrofluorometric analysis (484 nm excitation, 520 nm emission) using an automated reader (SpectraMax5e, Molecular Devices). For total cell uptake experiments, cell lysis was performed using lysis buffer (Pierce Chromatin Prep Module, Thermo Fisher Scientific, Switzerland) and the protein concentration of aliquots (10 µl per sample) of each lysate was determined by the Bradford method[37] (Bio-Rad Laboratories AG, Switzerland).

*Inductively coupled plasma-mass spectrometry measurements:* Prior to determination of ruthenium content, the chromatin extracts or cell lysates were digested with 400 µl of 69% nitric acid solution overnight at room temperature and adjusted with ultrapure water to a final volume of 4 ml. Metal concentrations were measured on an ICP-MS instrument (Elan DRC II, Perkin Elmer, Switzerland) equipped with a Meinhard nebulizer and a cyclonic spray chamber. The ICP-MS instrument was tuned daily using a solution provided by the manufacturer containing 1 p.p.b. each of Mg, In, Ce, Ba, Pb and U. $^{115}$In was used as an internal standard at a concentration of 0.5 p.p.b. External standards (ranging from 0.5 p.p.b. to 20 p.p.b.) were prepared gravimetrically in an identical matrix to the samples with single element standards (CPI International, Amsterdam, Holland). Three independent measurements were performed for each sample.

**Cell cycle analysis**. HeLa cells were seeded into 35 mm six-well plates, incubated overnight, and the following evening the agents at their respective IC$_{50}$ concentrations (Fig. 1) were added into triplicate samples for 40 h incubation. Subsequently, the medium containing compound was removed and the cells were collected for cell cycle analysis. Cells were rinsed in PBS, detached with trypsin, pelleted and washed before they were added in a dropwise manner into 70% ethanol. After overnight-incubation, cells were washed in PBS and incubated with DAPI dihydrochloride (3 µM, Life Technologies) for 15 min in the dark before analysing on a Fortessa flow cytometer (BD FACSDiva software, BD Biosciences). Analysis of cellular data was based on three independent replicates of each experiment. Graphic plots were generated with Excel (Microsoft Corporation). Mahalanobis multivariate analysis[38] was carried out using Xlstat (Addinsoft Corporation).

**DNA damage response analysis**. HeLa cells were seeded and incubated for 24 h before the complete media was replaced with either agent-free media or media containing either cisplatin, RAPTA-C, C10 or RR at their corresponding 72 h-IC$_{50}$ concentrations (cisplatin, $2.9 \pm 0.3$ µM; RAPTA-C, $603 \pm 179$ µM; C10, $5.8 \pm 1.8$ µM; RR, $2.3 \pm 0.4$ µM). Cells were grown for 72 h, then washed twice in PBS and harvested by scraping in ice-cold RIPA buffer containing inhibitors (2 mM PMSF, 8 mM NaF and 1× protease inhibitor cocktail (Roche)). The lysates were sonicated on ice, three pulses for 10 s on and 10 s off at 35% power. Samples and protein ladders (Cell Signaling) were run on 4–15% gradient gels (Mini-PROTEAN TGX cat. no. 456–1084), subsequently, they were wet-transferred onto 0.2 µm pore nitrocellulose (12369S Cell Signaling Technology) for 1.5 h at 70 V in transfer buffer containing 20% methanol. The blots were cut into strips and subjected to blocking with 5% nonfat milk in 1× Tris-buffered saline with Tween 20 detergent (TBST). Afterwards, each strip was incubated with the antibody corresponding to the size range of the protein of interest (Cell Signaling Technology (MA, USA) rabbit antibodies: phospho-Chk1 Ser345 (#2348), phospho-Chk2 Thr68 (#2197), phospho-histone H2A.X Ser139 (γH2A.X; #9718) and mouse anti-β-actin (#3700) as a loading control). Subsequently, they were incubated with the secondary anti-rabbit IgG horseradish peroxidase (HRP)-linked antibody or anti-mouse IgG HRP-linked antibody (#7074, #7076, respectively, Cell Signaling Technology) and the bands were visualized using the 3,3',5,5'-tetramethylbenzidine substrate for HRP (W4121 Promega, WI, USA). All antibodies were employed at a concentration corresponding to a 1000-fold dilution over the company-supplied aliquots. Uncropped western blot scans corresponding to the data shown in Supplementary Fig. 10 are given in Supplementary Fig. 11.

**Live cell imaging**. *Transfection of HeLa cells with H2B-EGFP and cell sorting:* HeLa cells were transfected with an H2B-EGFP plasmid[39] (plasmid #11680; obtained from Addgene, Cambridge, USA) using Lipofectamine 2000 reagent (Invitrogen, USA) according to the company's protocol. Stable transfectants were obtained by growing the cells in full growth media supplemented with 500 µg ml$^{-1}$ G418-sulfate (Promega). Subsequently, they were subjected to cell sorting; the cells were separated into high-expressing, medium-expressing and low-expressing populations (while the non-expressing cells were discarded). After growing up the different populations, they were tested for suitability for live imaging. Due to strong fluorescence and toxicity considerations during long exposure, the low-expressing population was selected for time-lapse experiments, after it was subjected to an additional round of sorting, where the non-expressing and high-expressing cells were eliminated.

*Time lapse confocal microscopy:* The H2B-EGFP-stably transfected and sorted HeLa cells were seeded in four out of eight wells of µ-slide eight-well ibiTreat dishes (80827, Ibidi, Germany). Cells were pre-treated for 16 h with the different drug or binuclear compounds (or agent-free control) and were then simultaneously imaged live for 24 h in complete growth DMEM medium without phenol red (31053-028, Gibco Life Technologies, USA) using a CorrSight confocal spinning disk microscope (FEI GmbH, Germany) equipped with an Orca R2 CCD camera (Hamamatsu, Japan) and a complete environmental control system (Ibidi GmbH) using the multi-stage position function in LA software (FEI GmbH). Time-lapse imaging was carried out at 37 °C, 5% CO$_2$ and 90% humidity in a closed-atmosphere chamber. Confocal z-stacks (1 µm z-interval; 20 µm total z-volume) were acquired with a 40× oil objective (NA 1.3, EC Plan Neofluar M27, Zeiss) every 10 min using the 488 nm laser line (65 mW; iChrome MLE-LFA) and standard GFP filter sets. Focus was maintained by a hardware autofocus system implemented in LA software, focus clamp. To minimize photo damage, the laser output power was set to 30–40% and a minimal exposure time (typically 100 ms) was chosen. Due to an additional and differential cytotoxic effect arising from sustained exposure to the blue laser light, drug and binuclear treatment concentrations were scaled to give a roughly equivalent number of cell death events over the course of imaging. This entailed 16 h pre-treatments with either 8 µM cisplatin, 40 µM RR, 100 µM C10, 600 µM C2, 600 µM PEG or 600 µM RAPTA-C.

**Nucleosome array studies**. *Array DNA production:* The nucleosome array DNA construct[19] consists of 36 repeats of the Widom 601 nucleosome core sequence that is flanked by 30 bp linkers (with the exception of the two termini), inserted into pUC18 vector. Plasmid was used to transform the DH5α strain of *Escherichia coli*, chosen to suppress recombination activities on the plasmid. One single colony was selected to inoculate 5 ml 2× TY media containing 100 µg ml$^{-1}$ ampicillin. The culture was incubated in a shaker incubator at 180 r.p.m., 37 °C, until the media appeared turbid (3–5 h) and transferred into 2 flasks, each containing 500 ml TB media with 100 µg ml$^{-1}$ Ampicillin, which were subsequently incubated at 150 r.p.m., 37 °C, for 16 h.

Cells were harvested by centrifugation at 8671×g, 20 °C, for 7 min and resuspended in 20 ml of lysis solution I (50 mM glucose, 10 mM EDTA, 25 mM Tris-HCl (pH 8.0)). Following addition of 50 mg lysozyme and incubation at room temperature for 15 min, 40 ml of lysis solution II (0.2 M NaOH, 1% w/v sodium dodecyl sulfate) was added. After a 5 min incubation on ice, 70 ml of lysis solution III (4 M K-acetate, 2 M acetic acid) was added. Precipitated cell debris was spun down at 9000×g, 4 °C, for 20 min and the supernatant was filtered through four layers of sterile gauze. Nucleic acid was precipitated by adding 0.6× volume of isopropanol, incubating on ice for 5 min, and centrifuging at 12,000×g, 4 °C, for 20 min. Pellet was dissolved in 10 ml TE(10,10) buffer (10 mM Tris-HCl [pH 8.0], 10 mM EDTA), and 10 ml of 5 M LiCl solution was added to precipitate the RNA and protein. After spinning down at 12,000×g, 20 °C, for 10 min, isopropanol precipitation was carried out as described above. Pellet was resuspended in 10 ml TE(10,1) buffer (10 mM Tris-HCl [pH 8.0], 1 mM EDTA), and 50 µl of 10 mg ml$^{-1}$ RNAse A was added. Following a 15 min incubation at room temperature, 0.2× volume of 4 M NaCl and 0.4× volume of 40% PEG 6000 were added. The mixture was incubated on ice for 30 min and centrifuged at 15,000×g, 4 °C, for 30 min. The pellet was then dissolved in 5 ml of TE(10,1) buffer, and 3 ml of phenol:chloroform (1:1 volume ratio) was added to separate plasmid from the remaining protein contaminants. Sample was centrifuged at 4000×g, 20 °C, for 10 min. The aqueous layer was collected and ethanol-precipitated. The purified plasmid pellet was dissolved in TE(10,1) buffer.

EcoRV digestion of the purified array plasmid was performed to release insert. Dra1 and HaeII enzymes were used to cut the vector down into smaller fragments. PEG precipitation was performed to remove most of the vector fragments. NaCl solution of 4 M and 40% PEG 6000 were added to the plasmid to yield final concentrations of 1–2 mg ml$^{-1}$ DNA, 0.54 M NaCl and 7% PEG 6000. After incubation on ice for 30 min, the mixture was centrifuged at 10,000×g for 30 min at 4 °C. The pellet was then dissolved in TE(10,1) and assessed by 10% polyacrylamide gel electrophoresis (PAGE). The PEG precipitation was then repeated, but with slightly less 40% PEG 6000 added to give a lower final concentration. After repeating this procedure two to three times, the sample would contain very little or no amount of digested vector fragment. Pure array DNA was ethanol-precipitated and stored in TE(10,1) buffer at −20 °C.

*Nucleosome array reconstitution:* The methodology for nucleosome array reconstitution was modified from refs. [19, 20, 40] and the online protocol (http://www.epigenesys.eu/images/stories/protocols/pdf/20111025120327_p42.pdf). Histone octamer (*H. sapiens*), in a solution of 10 mM triethanolamine (TEA)-hydrochloride [pH 7.5], 1 mM EDTA and 2 M NaCl, was mixed at different molar stoichiometry with array DNA (fixed at concentration of 0.1 mg ml$^{-1}$) to determine the optimal ratio for yielding saturated nucleosome array (Supplementary Fig. 12). Competitor DNA, consisting of 84/85 bp α-satellite DNA fragment, was added at a one-half mass-ratio with respect to the array DNA to prevent oversaturation of the array. A double-bag dialysis system was employed, whereby the sample in one dialysis bag was 'dialysed' against 10 mM TEA-HCl [pH 7.5], 1 mM EDTA and 2 M NaCl in a second dialysis bag, which was in turn dialysed against 10 mM TEA-HCl [pH 7.5] and 1 mM EDTA. In this way, the reconstitution kinetics were slowed, and the resulting quality of nucleosome array was consequently improved. Dialysis was carried out overnight at 4 °C, precipitate was removed by spinning down at 14,000×g, 4 °C, for 15 min, and the concentration of reconstituted nucleosome array was calculated based on the DNA concentration. The quality of the reconstituted nucleosome array was assessed by agarose gel electrophoresis (0.8% agarose with 0.2× TBE buffer). The saturation of the reconstituted nucleosome array was confirmed by digesting sample with AvaI enzyme (5000 units per mg DNA; cuts at the center of the linker DNA sections) at 37 °C for 1 h and then visualizing with 6% PAGE. When required, buffer exchange to 20 mM K-cacodylate [pH 6.0] was achieved using a 50 kDa-cutoff concentrator device.

*Electrophoretic mobility shift analysis:* For the derivatization studies, nucleosome array was maintained at a low, 13 nM, concentration to minimize precipitation and loss of material through inter-array cross-linking from the binuclear treatments. Nucleosome array in 20 mM K-cacodylate [pH 6.0] was incubated for 1 day at room temperature with a range of drug or binuclear concentrations of up to 46 µM for cisplatin and PEG and up to 230 µM for RAPTA-C and C2. Samples were subsequently passed through a DyeEx 2.0 spin column (Qiagen) to remove unbound metallo-compound before analysis by electrophoretic mobility shift analysis (0.8% agarose with 0.2× TBE buffer).

*Electron microscopy:* Samples of 13 nM nucleosome array in 20 mM K-cacodylate [pH 6.0] were incubated for 1 day at room temperature with 46 µM drug or binuclear agent. Samples were subsequently passed through a DyeEx 2.0 spin column (Qiagen) to remove unbound metallo-compound. Derivatized and native samples were diluted to attain 30 µg ml$^{-1}$ nucleosome array, which was subsequently dialysed against 10 mM TEA-HCl [pH 7.5], 40 mM NaCl buffer (with either 0, 0.5 or 1.6 mM MgCl$_2$) at 4 °C overnight. To fix the samples, 2 µl of 25% glutaraldehyde was added to 98 µl sample buffer to make a 0.5% stock. Two µl of this solution was then added to 8 µl of sample, so that the final glutaraldehyde concentration was 0.1%, followed by incubation on ice for 10 min.

Continuous carbon-coated grids were freshly prepared and glow-discharged before use[40]. Four microlitres of sample were deposited on grids for 1 min, blotted with filter paper and negatively stained with 1 drop of 2% (w/v) uranyl acetate solution. Electron microscopy images were collected on a T12 (FEI) transmission electron microscope equipped with a 4K Eagle ccd camera (FEI), at 120 keV and liquid nitrogen temperature. The nominal magnification was x49,000, calibrated object pixel size 0.2 nm and a defocus range between 1.0–2.0 µm.

**Nucleosome protein binding and cross-linking analysis**. *RCC1 protein production:* The gene coding for the *Drosophila* RCC1 protein inserted into the pST50Tr-STRaHISNdRCC1t1 plasmid[21] was a gift from S. Tan (Penn State, USA). The RCC1 gene was PCR-amplified and cloned into the pET28a bacterial expression vector, with a His-tag followed by a tobacco etch virus cleavage site at the N-terminus. RCC1 protein was overexpressed in *E. coli* BL21DE3 cells at reduced temperature, 18 °C, to maximize protein yields. The cells were harvested and resuspended in sonication buffer (20 mM Tris-HCl [pH 7.5], 5% glycerol, 500 mM NaCl, 0.5 mM phenylmethylsulfonyl fluoride and 0.05% (v/v) 1× protease inhibitor cocktail (Roche)) and then lysed using a homogenizer. The lysate was clarified by centrifugation at 50,000×g, and the supernatant was loaded onto a 5 ml IMAC-Ni column pre-equilibrated with 20 mM Tris-HCl [pH 7.5], 5% glycerol, 500 mM NaCl buffer. The protein was eluted with an imidazole gradient and subjected to Tobacco Etch Virus (TEV) protease cleavage. After complete TEV digestion, the sample was passed over an IMAC-Ni column to remove the His-tag. The flow-through, containing RCC1, was further purified using a diethylaminoethyl sepharose fast flow (DEAE FF) ion exchange column (GE Healthcare Life Sciences). Fractions containing RCC1 protein were pooled together, concentrated to 8 mg ml$^{-1}$ and stored at −80 °C.

*Protein binding and cross-linking assays:* RCC1 binding and cross-linking analyses were conducted using NCP assembled with *H. sapiens* histones and a 145 bp DNA fragment[32]. For the RCC1 experiments, 1 µM NCP, in a buffer of 20 mM K-cacodylate [pH 6.0], was incubated with 10/20 µM RAPTA-C (Fig. 8a), 100/150 µM RAPTA-C (Fig. 8b), 10/15/20 µM RR, 5/10 µM C2 or 5/10 µM C10 (Fig. 8a) at room temperature for ~13 h. Native or treated NCP (0.5 µM) were subsequently incubated with RCC1 (0.5 µM) in 20 mM K-cacodylate [pH 6.0] buffer on ice for 15 min. The samples were subjected to native PAGE at 4 °C, and gels were stained with coomassie brilliant blue. For the nucleosome cross-linking experiments, 1 µM NCP, in a buffer of 20 mM K-cacodylate [pH 6.0], was incubated with 10/20 µM RAPTA-C, 10/20 µM RR, 10/20 µM C2 or 10/20 µM C10 at room temperature for 24 h. Samples were subsequently analysed with native PAGE, and gels were stained with coomassie brilliant blue.

For the analysis of histone protein cross-linking under denaturing conditions, 1 μM NCP, in a buffer of 20 mM K-cacodylate [pH 6.0], was incubated with 10/20 μM RAPTA-C, 10/20 μM RR, 10/20 μM C2 or 10/20 μM C10 at room temperature overnight. Five microlitres of 5× loading dye (250 mM Tris-HCl [pH 6.8], 10% w/v sodium dodecyl sulfate (SDS), 25 mM dithiothreitol, 0.1% w/v bromophenol blue, 50% v/v glycerol) was added to the 20 μl reaction mix, followed by a 1 min incubation at 95 °C. After brief centrifugation, samples were loaded onto a 15% SDS-PAGE gel, and the gel was subsequently stained with coomassie brilliant blue. Histone protein identity was assessed by subjecting bands excised from the SDS-PAGE gel to matrix assisted laser desorption/ionization-time of flight (MALDI-TOF) mass spectrometry and by western blot analysis. Anti-H2A (ab13923) and anti-H2B (ab1790) western blotting was performed following the vendor's protocol (AbCam, UK; primary antibodies employed at 1000-fold dilution relative to company-supplied aliquots). The secondary antibody corresponded to HRP-linked goat pAb anti-rabbit IgG (ab6721) employed at 5000-fold dilution. Data shown in Supplementary Fig. 14b correspond to uncropped blot scans.

**Data availability**. Atomic coordinates and structure factors for the NCP models with RR, SS, RS and C2 adducts have been deposited in the Protein Data Bank under accession codes 5XF3, 5XF4, 5XF5 and 5XF6, respectively. Other data that support the findings of this study are available from the corresponding authors upon request.

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

## Acknowledgements

We thank M. Wang, V. Olieric and staff at the Swiss Light Source (Paul Scherrer Institute, Villigen, Switzerland). We thank H. Kurumizaka and T.S. Kumaravel for providing us with the human core histone expression plasmids, S. Tan for the RCC1 plasmid, P. Dröge and C. Battistella for HeLa cells, A. Routh for cloning the nucleosome array and A. Wong for help with EM imaging. The H2B-EGFP plasmid was a gift from G. Wahl. We are grateful for financial support by the Singapore Ministry of Education Academic Research Fund Tier 3 Programme (grant MOE2012-T3-1-001), Ministry of Health National Medical Research Council (grant NMRC/1312/2011) and the NTU Institute of Structural Biology. This research was also supported by the Swiss National Science Foundation via individual grant no. 200020-140865, the NCCR in Chemical Biology and by start-up funding from the University of Hull (UK).

## Author contributions

G.E.D. conducted the cell growth inhibition and cell cycle assays, confocal time lapse microscopy, DNA damage marker tests and contributed to manuscript writing; Z.A. carried out the electrophoretic and EM nucleosome array studies and contributed to the crystal structure work; Z.M. conducted the X-ray crystallographic analysis and contributed to the electrophoretic analysis of array; T.R. performed the cellular uptake and chromatin-binding studies; D.S. carried out the nucleosome cross-linking and RCC1-binding analyses; S.P. set up our human core histone production platform and produced RCC1 protein; D.R. provided the nucleosome array construct and expertise for producing the array; A.L. provided expertise for confocal microscopy; S.S. provided expertise for the

EM analysis; B.S.M. and P.J.D. designed and synthesized the binuclear compounds; C.A. D. designed the study, conducted data analysis and wrote the manuscript.

## Additional information

**Competing interests:** The authors declare no competing financial interests.

