## [Peer Review File · Nature Communications]

Reviewers' comments:

Reviewer #1 (Remarks to the Author):

Herein, Davey et al. describe a family of binuclear ruthenium compounds that selectively target the nucleosome acidic patch. This is elegant study that after some modifications could be published in Nature Communications.

This reviewer has the following comments:

1. The compounds described in this manuscript operate via a new mechanism of action, completely different to that of cisplatin and other classical platinum drugs. The compounds were also designed to target the chromatin acidic patch, rather than being a chance discovery, and in these senses the work is highly original and interesting and should be of interest to both medicinal inorganic chemists and those working on chromatin. I could imagine that these compounds could easily become tools widely used by those in the field who want to block the acidic patch of chromatin to study the resulting consequences.

2. The compounds described in this manuscript operate via a new mechanism of action, completely different to that of cisplatin and other classical platinum drugs. The compounds were also designed to target the chromatin acidic patch, rather than being a chance discovery, and in these senses the work is highly original and interesting and should be of interest to both medicinal inorganic chemists and those working on chromatin. I could imagine that these compounds could easily become tools widely used by those in the field who want to block the acidic patch of chromatin to study the resulting consequences.

3. The authors make comparisons to cisplatin several times, and it would be good to be a bit more quantitative and indicate the amounts of cisplatin and the dinuclear compounds that bond to their established targets, i.e. DNA and chromatin, respectively. While I suggest adding this data, and appreciate that all those working on metal drugs compare their results to cisplatin, the RAPTA compounds and these new RAPTA-like dinuclear compounds are very different from cisplatin, so comparisons with other non-metal drugs are equally as valid. I suggest therefore adding some discussion on other compounds that bind to the acidic patch of chromatin from the literature.

4. Could the results described in this manuscript explain the observations described in the following article on RAPTA-C: *Sci. Rep.*, 2017, 7, 43005? It seems to me that there is a correlation and it might be interesting to comment on this in the discussion since it would help bridge the in vitro results with in vivo effects described in this paper.

5. The RAPTA compound in this figure and elsewhere, e.g. Fig 4 must be defined, presumably it is RAPTA-C (as marked in abstract?).

6. For cosmetic reasons I suggest embedding the scale bar within the figure and removing the black section at the bottom of each image.

Reviewer #2 (Remarks to the Author):

Ruthenium compounds are of interest because they may be effective alternatives to platinum anti-cancer drugs. The authors have shown previously that, unlike cisplatin, some Ru compounds target the histones responsible for DNA packaging rather than the DNA itself and are less cytotoxic than those that create DNA lesions. In this study, the authors describe the synthesis and characterisation of a novel set of binuclear Ru compounds that are capable of protein cross-linking. These compounds have two Ru²⁺ ions bound in RAPTA-C-like moieties separated by different linkers: PEG, C10 and C2 have flexible linkers whereas RR, RS and SS have rigid linkers (the

nomenclature refers to three different stereoisomers). They compare the properties of these compounds with those of cisplatin and RAPTA-C (which is essentially a mono-nuclear control). By crystallography, they show that all of the binuclear compounds form adducts with histone residues in the acidic patch on the surface of the nucleosome. The acidic patch includes H2A-E61/E64 and H2B-E102/H106, and is often recognised by nucleosome-binding proteins such as RCC1. Other in vitro data (electrophoretic mobility shift assays and electron microscopy of reconstituted nucleosome fibres) indicate that these adducts may be intra-nucleosomal (between H2A and H2B acidic patch residues) or inter-nucleosomal. In addition, they show that RCC1 binding to the acidic patch is blocked by these drugs, consistent with targeting of the compounds to the acidic patch. These binuclear Ru reagents do not activate the DNA damage checkpoint in HeLa cells, consistent with a mechanism involving histone adducts rather than DNA damage. Instead, the drugs induce condensation of the cellular chromatin, eventually followed by apoptosis, involving a substantially different mechanism from that of cisplatin.

This is an important and interesting study of a set of novel compounds with unusual properties. The experiments are performed well and the results are convincing. I have the following minor comments:

1. In Fig. 7, addition of the PEG compound results in condensation of a reconstituted chromatin fibre containing 36x 177 bp-repeats even in the absence of magnesium, most likely due to inter-nucleosome cross-links. However, can the authors rule out the possibility that the condensation is due to DNA charge neutralisation by the binuclear compound, which has a charge of +4? That is, PEG may behave like spermine, which also has a charge of +4?
2. Are the Ru-adducts resistant to SDS? If so, histone-histone cross-links in mono-nucleosomes and reconstituted chromatin fibres could be demonstrated directly in a protein gel.
3. The titration in Fig. S12 ends with 2 histone octamers/nucleosome - is this correct? More explanation is needed in the legend.

Reviewer #3 (Remarks to the Author):

This paper represents the biological interactions of some ruthenium complexes which are closely related to already published molecules. The science is interesting, but the paper has been poorly put together. For example, the first two lines of the abstract do not make much sense. There are some interesting results in the paper, that are more suited to a specialised Journal. Therefore, I recommend rejection, rewriting and publication in a more suitable journal.

RESPONSES TO REVIEWER COMMENTS

We thank the reviewers wholeheartedly for their valuable time and input on our manuscript. We have considered all of the recommendations of the reviewers, whose insightful comments have been very helpful in compiling this revised version. Below we outline our responses to the specific points of each reviewer in turn.

In response to comments by Reviewer #1:

1. *The authors make comparisons to cisplatin several times, and it would be good to be a bit more quantitative and indicate the amounts of cisplatin and the dinuclear compounds that bond to their established targets, i.e. DNA and chromatin, respectively. While I suggest adding this data, and appreciate that all those working on metal drugs compare their results to cisplatin, the RAPTA compounds and these new RAPTA-like dinuclear compounds are very different from cisplatin, so comparisons with other non-metal drugs are equally as valid. I suggest therefore adding some discussion on other compounds that bind to the acidic patch of chromatin from the literature.*

We have extended and reanalysed the ICP-MS data to allow for a direct comparison of the cell uptake and chromatin binding values. This correspondingly provides an estimate of the chromatin targeting proclivity for the binuclears and RAPTA-C. The second paragraph of the “Generate Substantial Chromatin Adducts in Cells” subsection in Results has been revised and a new paragraph (now second) in the Discussion section added. Also, Figure 3 has been modified and expanded to include the previous Figure S9 data (Fig. S9 accordingly removed). As such, there is now a full discussion of the localization attributes, including comparison to findings on cisplatin (new ref. 22). In addition, we have included further discussion of other known acidic patch-binding compounds (last paragraph in Discussion, including new ref. 28).

2. *Could the results described in this manuscript explain the observations described in the following article on RAPTA-C: Sci. Rep., 2017, 7, 43005? It seems to me that there is a correlation and it might be interesting to comment on this in the discussion since it would help bridge the in vitro results with in vivo effects described in this paper.*

Indeed this can help rationalize our findings on RAPTA-C, and this has been added to the Discussion (paragraph 2, including new ref. 23).

3. *The RAPTA compound in this figure and elsewhere, e.g. Fig 4 must be defined, presumably it is RAPTA-C (as marked in abstract?).*

This has now been clarified throughout all of the figures and the text.

4. *For cosmetic reasons I suggest embedding the scale bar within the figure and removing the black section at the bottom of each image.*

This has been done for Figures 7 and S12 (previously S13).

In response to comments by Reviewer #2:

5. *In Fig. 7, addition of the PEG compound results in condensation of a reconstituted chromatin fibre containing 36x 177 bp-repeats even in the absence of magnesium, most likely due to inter-nucleosome cross-links. However, can the authors rule out the possibility that the condensation is due to DNA charge neutralisation by the binuclear compound, which has a charge of +4? That is, PEG may behave like spermine, which also has a charge of +4?*

Although electrostatics must be involved, it is unlikely that the effect is purely electrostatic, since the acidic patch-targeting LANA peptide, which also carries the same 4+ charge, does not display the same robust chromatin condensing activity as the binuclears. We have added further discussion of this in the second to last paragraph of the Discussion.

6. *Are the Ru-adducts resistant to SDS? If so, histone-histone cross-links in mononucleosomes and reconstituted chromatin fibres could be demonstrated directly in a protein gel.*

Although the harsh conditions required to denature the nucleosome are likely to also alter the ruthenium adducts (to at least some extent), we conducted a number of trials on RAPTA-C and the binuclears with denaturing electrophoretic gel analysis, which does indeed indicate that distinct cross-linked histone species are formed by the binuclears. This is now described in the Results section (“Impede Protein Binding & Cross-Link Nucleosomes” subsection), with a new Figure, S13, added to the SI.

7. *The titration in Fig. S12 ends with 2 histone octamers/nucleosome - is this correct? More explanation is needed in the legend.*

Initial titrations were conducted to cover a very broad stoichiometric range, in order to ensure that saturation of the array could be achieved (confirmed later by *Ava*I restriction enzyme digestion; described in Methods). We have added an explanation of this in the legend of Figure S11 (previously S12).

In response to comments by Reviewer #3:

8. *This paper represents the biological interactions of some ruthenium complexes which are closely related to already published molecules. The science is interesting, but the paper has been poorly put together. For example, the first two lines of the abstract do not make much sense. There are some interesting results in the paper, that are more suited to a specialised Journal. Therefore, I recommend rejection, rewriting and publication in a more suitable journal.*

The novel binuclear metal compounds we have studied here show unusual activities which, to our knowledge, have not been observed previously for any other types of compounds. We have composed the manuscript as carefully as possible with input from all of the co-authors. To ensure clarity, we have rewritten the first two sentences of the abstract.

REVIEWERS' COMMENTS:

Reviewer #1 (Remarks to the Author):

The authors significantly improved the quality of the paper and answered convincingly to all the points I've mentioned in my review. I do not have any further comments and the paper might be considered for the publication in Nature Communications.

Reviewer #2 (Remarks to the Author):

The revised manuscript addresses my concerns. This is a very nice, interesting paper.